# Resistance patterns and clinical outcomes of *Klebsiella pneumoniae* and invasive *Klebsiella variicola* in trauma patients

John L. Kiley[1]*, Katrin Mende[1,2,3], Miriam L. Beckius[1], Susan J. Kaiser[1,2,3], M. Leigh Carson[2,3], Dan Lu[2,3], Timothy J. Whitman[4¤a], Joseph L. Petfield[5¤b], David R. Tribble[2], Dana M. Blyth[1¤c]

1 Brooke Army Medical Center, JBSA Fort Sam Houston, San Antonio, Texas, United States of America, 2 Department of Preventive Medicine and Biostatistics, Infectious Disease Clinical Research Program, Uniformed Services University of the Health Sciences, Bethesda, Maryland, United States of America, 3 Henry M. Jackson Foundation for the Advancement of Military Medicine, Inc., Bethesda, Maryland, United States of America, 4 Walter Reed National Military Medical Center, Bethesda, Maryland, United States of America, 5 Landstuhl Regional Medical Center, Landstuhl, Germany

¤a Current address: University of Vermont Medical Center, Burlington, Vermont, United States of America
¤b Current address: Nemours Alfred I. duPont Hospital for Children, Wilmington, Delaware, United States of America
¤c Current address: Walter Reed National Military Medical Center, Bethesda, Maryland, United States of America
* john.l.kiley.mil@mail.mil

**Data Availability Statement:** All relevant data are within the manuscript and its Supporting Information files.

**Funding:** Support for this work (IDCRP-024) was provided by the Infectious Disease Clinical

## Abstract

Recent reclassification of the *Klebsiella* genus to include *Klebsiella variicola*, and its association with bacteremia and mortality, has raised concerns. We examined *Klebsiella* spp. infections among battlefield trauma patients, including occurrence of invasive *K. variicola* disease. *Klebsiella* isolates collected from 51 wounded military personnel (2009–2014) through the Trauma Infectious Disease Outcomes Study were examined using polymerase chain reaction (PCR) and pulsed-field gel electrophoresis. *K. variicola* isolates were evaluated for hypermucoviscosity phenotype by the string test. Patients were severely injured, largely from blast injuries, and all received antibiotics prior to *Klebsiella* isolation. Multidrug-resistant *Klebsiella* isolates were identified in 23 (45%) patients; however, there were no significant differences when patients with and without multidrug-resistant *Klebsiella* were compared. A total of 237 isolates initially identified as *K. pneumoniae* were analyzed, with 141 clinical isolates associated with infections (remaining were colonizing isolates collected through surveillance groin swabs). Using PCR sequencing, 221 (93%) isolates were confirmed as *K. pneumoniae*, 10 (4%) were *K. variicola*, and 6 (3%) were *K. quasipneumoniae*. Five *K. variicola* isolates were associated with infections. Compared to *K. pneumoniae*, infecting *K. variicola* isolates were more likely to be from blood (4/5 versus 24/134, p = 0.04), and less likely to be multidrug-resistant (0/5 versus 99/134, p<0.01). No *K. variicola* isolates demonstrated the hypermucoviscosity phenotype. Although *K. variicola* isolates were frequently isolated from bloodstream infections, they were less likely to be multidrug-resistant. Further work is needed to facilitate diagnosis of *K. variicola* and clarify its clinical significance in larger prospective studies.

Research Program (IDCRP), a Department of Defense program executed through the Uniformed Services University of the Health Sciences, Department of Preventive Medicine and Biostatistics through a cooperative agreement with The Henry M. Jackson Foundation for the Advancement of Military Medicine, Inc. (HJF). This project has been funded by the National Institute of Allergy and Infectious Diseases, National Institutes of Health, https://www.niaid.nih.gov/, under Inter-Agency Agreement Y1-AI-5072 to DRT, the Defense Health Program, U.S. DoD, under award HU0001190002 to DRT, the Department of the Navy under the Wounded, Ill, and Injured Program (HU0001-10-1-0014) to DRT, and the Military Infectious Diseases Research Program, https://midrp.amedd.army.mil/ (HU0001-15-2-0045) to KM. The funders had no role in study design, data collection and analysis, decision to publish, or preparation of the manuscript. Support in the form of salaries was provided by HJF for authors KM, SJK, MLC, and DL; HJF did not have any additional role in the study design, data collection and analysis, decision to publish, or preparation of the manuscript. The specific roles of these authors are articulated in the 'author contributions' section.

**Competing interests:** KM, SJK, MLC, and DL are employees of the Henry M. Jackson Foundation for the Advancement of Military Medicine, Inc. (HJF), a not-for-profit Foundation authorized by Congress to support research at the Uniformed Services University of the Health Sciences (USU) and throughout military medicine. This does not alter our adherence to PLOS ONE policies on sharing data and materials. Please see Data Availability Statement.

## Introduction

*Klebsiella pneumoniae* is recognized as a frequent cause of healthcare-associated infections, including bloodstream infections (BSI), urinary tract infections, ventilator-associated pneumonia, and surgical site infections [1]. Among military personnel with battlefield injuries sustained in Iraq and Afghanistan, *K. pneumoniae* was one of the most common colonizing Gram-negative bacilli identified from groin surveillance cultures collected at hospital admission, with approximately 22% of isolates being classified as multidrug-resistant (MDR) [2,3]. Furthermore, *K. pneumoniae* was the third most frequently identified isolate from wounded warriors with MDR Gram-negative bacilli infections [4].

Re-classification of the *Klebsiella* genus and description of new *Klebsiella* species has raised questions regarding species specific virulence [5–9]. Certain themes have emerged from early research done after this re-classification. Most notably, there is an association between *Klebsiella variicola* and healthcare-associated infections with potential for worsened outcomes and greater invasive disease (e.g., bacteremia) when compared to *K. pneumoniae* [5–10]. However, most modern clinical laboratories misidentify *K. variicola* as *K. pneumoniae* and only recently has polymerase chain reaction (PCR) and mass spectrometry techniques for identification been described [11–13]. Thus, epidemiology, clinical manifestations, and virulence patterns of *K. variicola* remain controversial.

With the importance of *K. variicola* as an emerging pathogen, we evaluated *Klebsiella* infections in patients who suffered battlefield-related trauma [8]. Specifically, we examined the epidemiology and resistance patterns of *Klebsiella* spp. infections, identified prior misclassifications of *Klebsiella* species, and assessed the prevalence and incidence of invasive *K. variicola* disease in this population.

## Materials and methods

### Study population and definitions

Data were collected through the Trauma Infectious Disease Outcomes Study (TIDOS), which is a retrospective observational study of infectious complications among military personnel wounded in Iraq or Afghanistan (2009–2014) [14,15]. All patients in the TIDOS population were ≥18 years of age active-duty personnel or Department of Defense (DoD) beneficiaries who initially received care in the combat theater, followed by medical evacuation to Landstuhl Regional Medical Center (LRMC) in Germany with ultimate transfer to participating military hospitals in the United States. The participating U.S. military hospitals were Brooke Army Medical Center in San Antonio, TX, and Walter Reed National Military Medical Center in the National Capital Region (National Naval Medical Center and Walter Reed Army Medical Center prior to September 2011) [14].

The Institutional Review Board (IRB) of the Uniformed Services University (Bethesda, MD) approved this study. Data were collected from subjects that provided authorization for the collection and analysis of their data through informed consent and HIPAA authorization processes, or through an IRB-approved waiver of consent for use of de-identified data not obtained through interaction or intervention with human subjects.

Inclusion in this analysis required isolation of a *Klebsiella* spp. isolate associated with clinical diagnosis of infection. Demographics, trauma characteristics, and information on casualty care were collected from the DoD Trauma Registry (DoDTR) [16] and infection-related information (e.g., infection syndromes, microbiology, and antibiotic treatment) was obtained from the TIDOS Infectious Disease module of the DoDTR. Infectious disease events were identified using a combination of clinical and laboratory findings and classified using National

Healthcare Safety Network definitions as previously described [14,17]. Colonization was defined as recovery of isolates from groin swabs obtained as part of targeted infection control surveillance at hospital admission for their deployment-related injury. All other isolates were defined as infecting isolates as they were recovered during workups for clinical infection. Multidrug resistance was defined using Centers for Disease Control and Prevention criteria as either resistance to ≥3 classes of aminoglycosides, β-lactams, carbapenems, and/or fluoroquinolones or production of an extended spectrum beta-lactamase (ESBL) or carbapenamase [18].

### *Klebsiella* spp. isolate analysis

Initial identification of organisms was performed by participating hospitals' clinical microbiology laboratories. All hospital labs used either BD Automated Microbiology System (BD Diagnostics, Sparks, MD) or Vitek 2 (bioMérieux Inc., Hazelwood, MO). These two platforms are only able to identify *K. pneumoniae* and not differentiate this identification from *K. variicola*, or other members of the *Klebsiella* complex [9]. Isolates were stored at -80˚C in a central TIDOS specimen repository.

All initial and serial infecting isolates identified as *K. pneumoniae* and stored in the TIDOS specimen repository were included for analysis. Serial isolation was defined as an isolate collected ≥7 days from a prior isolate. All colonizing isolates linked with infecting isolates (defined as isolation from groin admission swab prior to infection) were included. A convenience sample of the remaining colonizing isolates (50 MDR and 50 non-MDR colonizing *K. pneumoniae* archived isolates) were chosen randomly from the repository.

All isolates underwent passage on 5% sheep blood agar twice prior to confirmatory identification and antimicrobial susceptibility testing utilizing BD Phoenix Gram-negative panel (NMIC/ID-304) and BD Automated Microbiology System (BD Diagnostics, Sparks, MD)–this technique replicates typical clinical laboratory procedures and is only able to identify *K. pneumoniae*. Breakpoints were determined using Clinical Laboratory Standards Instituted guidelines (M100, 28th edition, 2018). Antimicrobials tested included cefazolin, ceftriaxone, cefepime, levofloxacin, amoxicillin-clavulanate, piperacillin-tazobactam, aztreonam, meropenem, ertapenem, amikacin, and trimethoprim/sulfamethoxazole.

All isolates underwent DNA extraction using QIAamp DNA Mini Kit (QIAGEN, Hilden, Germany) and subsequently pulsed-field gel electrophoresis (PFGE) analysis to assess for clonality. Any uninterpretable gel patterns on PFGE were repeated. In order to identify *K. pneumoniae*, *K. quasipneumoniae*, and *K. variicola*, all extracted DNA samples underwent PCR using the method and oligonucleotide primer sequences described by Garza-Ramos and colleagues [12,19]. Of note, this work was completed prior to the extensive molecular epidemiologic work resulting in the further expansion of the *Klebsiella pneumoniae* complex [8,9]. Due to early case reports of *K. variicola* being linked with hypermucoviscosity phenotypes, string tests were performed on all *K. variicola* isolates. String test length for positivity was defined as stranding from one colony >5 millimeters from the agar surface. We did not perform the string test on *K. pneumoniae* or *K. quasipneumoniae* isolates as our focus was describing any association between the hypermucoviscous phenotype and *K. variicola*.

### Statistical analysis

All patients with isolation of *Klebsiella* spp. were initially analyzed as a group. Patients who were subsequently identified to have *K. variicola* isolation by PCR assay were evaluated in a secondary analysis to examine clinical predictors of isolation and phenotypic behaviors. Univariate analysis by $X^2$ and Fisher's Exact Test was performed for categorical variables where appropriate. Continuous variables were analyzed using Mann-Whitney U. Statistical analysis

was performed using IBM SPSS Statistics 22 (Version 22 IBM, NY, 2013). A *p* value of $<0.05$ was considered significant. Data availability: All relevant data are provided within the paper and its supporting documentation.

## Results

### Study population

Among 2,699 TIDOS patients, 51 patients with infecting *Klebsiella* isolates met inclusion criteria for the analysis. All patients were young men with a median age of 23 years who were severely injured with 82% sustaining blast trauma, largely from improvised explosive devices (Table 1). All 51 patients received antibiotics prior to recovery of infecting *Klebsiella* spp. isolates. The most common antibiotics administered prior to recovery of an infecting isolate were tetracyclines (N = 46, 90%), first generation cephalosporins (N = 45, 88%), and vancomycin (N = 39, 76%). Duration of hospitalization was a median of 49 days. Overall, four (8%) patients died.

Sources of the initial 51 infecting *Klebsiella* isolates were respiratory (N = 16, 31%), wound (N = 13, 26%), blood (N = 10, 20%), urine (N = 5, 10%), intra-abdominal (N = 4, 8%), and other (N = 3, 6%). There were a median of 23 (interquartile range [IQR]: 22–55) days between isolation of initial infecting isolate and death for the four patients who died.

Twenty-three patients (45%) had initial *Klebsiella* spp. infections that were MDR. When compared to patients with initial non-MDR *Klebsiella* spp. infections (N = 28, 55%), there was no significant difference in age (median of 22 years [IQR: 21–29] with MDR infections versus 22 years [IQR: 21–26] for non-MDR infections, p = 0.42), or duration between injury and isolation of first infecting isolate (median of 10 [IQR: 7–27] days versus 19 [IQR: 9–37] days, p = 0.25). In addition, injury severity was similar between the two groups with patients who had MDR *Klebsiella* infections having a median injury severity score (ISS) of 38 (IQR: 32–46) compared to a median of 35 (IQR: 28–45; p = 0.65) for patients with non-MDR *Klebsiella* infections.

Sixteen (31% of 51) patients had serial isolation of *Klebsiella* spp. with sources being respiratory (N = 7, 43%), wound (N = 4, 25%), blood (N = 3, 19%), and urine (N = 2, 13%). Age, ISS, and days between injury and first infecting isolate of patients who had serial isolation of any *Klebsiella* spp. and those without (N = 35, 69%) were not statistically significant; however, patients with serial *Klebsiella* isolation trended towards higher mortality (19%) compared with patients who only had initial isolation (3%, p = 0.07; Table 2).

### Pulsed-field gel electrophoresis patterns

There were 12 unique PFGE patterns/types (PFTs) that were each identified in >1 patients with PFT 78 recovered from ten unique patients and PFT 80 from three unique patients. The remaining 10 PFTs were identified in two patients each. Seven (58% of 12) PFTs were associated with infections; however, none of the strains were isolated from patients who were treated at the same initial facilities. There were five (42%) unique PFTs identified among colonizing isolates recovered from patients who were evacuated from the same initial facility (Bastion, Afghanistan). Only one of these PFTs (PFT 78; identified from five colonized-patients) was identical to the PFT of an infecting isolate from a single patient (Table 3). The infecting isolate was recovered from a wound infection eight days after the first colonizing isolate was collected from the other patients. Isolation of this strain with PFT 78 from patients evacuated from Bastion ended four months after the first isolate was recovered. All infecting and colonizing isolates corresponding to PFT 78 were MDR.

**Table 1. Characteristics of patients with *Klebsiella* species infections.**

| Characteristic or outcome, No. (%) | Patients with *Klebsiella* spp. isolates N = 51 |
|---|---|
| Age, years, median (IQR) | 23 (21–28) |
| Male sex | 51 (100) |
| Injury severity score, median (IQR) | 38 (15–45) |
| *Injury Mechanism* | |
| Blast injury | 42 (82) |
| Improvised explosive device | 37 (72) |
| Gunshot wound | 6 (12) |
| *Initial facility geographic location[a]* | |
| **Southern Afghanistan:** | **34 (67)** |
| Bastion | 13 (25) |
| Role 2a | 1 (2) |
| Kandahar | 18 (35) |
| Role 2b | 1 (2) |
| Role 2c | 1 (2) |
| **Eastern/Northeastern Afghanistan:** | **10 (20)** |
| Role 2d | 1 (2) |
| Role 2e | 1 (2) |
| Role 2f | 3 (6) |
| Role 2g | 1 (2) |
| Role 2h | 1 (2) |
| Role 2i | 2 (4) |
| Role 2j | 1 (2) |
| **Central Afghanistan:** | **2 (4)** |
| Bagram | 1 (2) |
| Role 2k | 1 (2) |
| **Iraq** | **1 (2)** |
| **Landstuhl Regional Medical Center (Germany)** | **2 (4)** |
| **Other** | **2 (4)** |
| *U.S. military hospital* | |
| Brooke Army Medical Center | 24 (47) |
| National Capital Region | 26 (51) |
| Use of Mechanical ventilation | 42 (82) |
| Antibiotic exposure prior to isolation of *Klebsiella* spp. | 51 (100) |
| Days between injury and 1st infecting isolate, median (IQR) | 15 (8–33) |
| Total length of hospital stay, median days (IQR) | 49 (28–70) |
| Death | 4 (8) |

IQR–interquartile range.

[a] Role 2 facilities are within the operational theater with a tent or structure-based operating room and limited personnel (mobile forward surgical teams for initial and resuscitative care are included). Bastion, Kandahar, and Bagram are Role 3 facilities, which are combat support hospitals within the operational theater.

## Isolate analysis

Two hundred and thirty-seven isolates from 121 patients included in the study were initially identified as *K. pneumoniae*, of which 141 were infecting isolates and 96 were colonizing isolates (four colonizing isolates initially chosen were excluded after discovering they were identified incorrectly as *Klebsiella* species). After undergoing PCR, 221 (93% [95% confidence

**Table 2. Patients with serial isolation of *Klebsiella* spp. versus those with initial isolation only.**

| Characteristic, median (IQR) | Patients with single isolates (N = 35) | Patients with serial isolation (N = 16) | P-value |
|---|---|---|---|
| Age, years | 22 (21–28) | 22 (21–28) | 0.23 |
| Injury Severity Score | 37 (30–45) | 28 (30–45) | 0.42 |
| Days between injury and first infecting isolate | 16 (8–34) | 15 (8–33) | 0.81 |
| Length of hospital stay, days | 41 (29–59) | 62 (26–80) | 0.41 |
| Death, No (%) | 1 (3) | 3 (19) | 0.07 |

IQR–interquartile range.

interval [CI]: 90–96]) isolates were verified as *K. pneumoniae* (from 108 patients; 134 infecting isolates, 87 colonizing isolates from groin swabs), 10 (4% [95% CI: 1.5–6.5]) were identified as *K. variicola* (from 8 patients; 5 infecting isolates, 5 colonizing isolates), and 6 (3% [95% CI: 0.8–5.2]) as *K. quasipneumoniae* (from 5 patients; 2 infecting isolates, 4 colonizing isolates). The years of *K. pneumoniae* collection were 28 (13% of 221) isolates in 2009, 95 (43%) in 2010, 30 (14%) in 2011, 51 (23%) in 2012, 13 (6%) in 2013, and 4 (2%) in 2014. There were no *K. variicola* isolates collected in 2009, while three were collected in 2010, four were collected in 2011, and one isolate was collected per year from 2012–2014. Four of the *K. quasipneumoniae* isolates were collected in 2011, one isolate in 2013, and one isolate in 2014.

Sources of the 134 *K. pneumoniae* infecting isolates were wound (N = 57, 42%), respiratory (N = 29, 22%), blood (N = 24, 18%), intraabdominal (N = 4, 3%), and other (N = 20, 15%). Of the 68 initial infecting *Klebsiella* spp. isolates, 43 were MDR (63%). Similarly, 11 (65%) of 17 colonizing isolates obtained prior to infecting isolates were MDR. Antimicrobial susceptibility patterns demonstrated substantial resistance to cephalosporins, fluoroquinolones, and piperacillin/tazobactam (Table 4). Five infecting isolates recovered after non-MDR colonizing isolates from the same patient demonstrated resistance to new antimicrobials: two developed resistance to cefazolin at 10 and 17 days after colonization, one to piperacillin/tazobactam at 19 days, one to ertapenem at 7 days, and one to gentamicin at 200 days after colonization.

## *Klebsiella variicola*

There were 10 (4% of 237 *Klebsiella* isolates) isolates of *K. variicola* identified from eight patients (7%). Five isolates were colonizers from groin swabs, while the five isolates associated with infections were from blood (N = 4), and intraabdominal specimens (N = 1). Among 51 patients with infections, 48 (94%) had infections attributed to *K. pneumoniae* and 3 (6%) had *K. variicola* infections. The three patients with *K. variicola* infections had a median age of 25

**Table 3. Outbreak analysis of the single pulsed-field gel electrophoresis type (PFT) 78.**

| Patient | PFT | Initial Location | Date of isolation | Source of initial clonal isolates | |
|---|---|---|---|---|---|
| | | | | Wound (infecting) | Groin (colonizing) |
| I | 78 | Bastion | 16 August 2010 | 0 | 1 |
| J | 78 | Bastion | 16 August 2010 | 0 | 1 |
| K | 78 | Bastion | 21 August 2010 | 0 | 1 |
| L | 78 | Bastion | 23 August 2010 | 0 | 1 |
| M | 78 | Bastion | 24 August 2010 | 1 | 0 |
| N | 78 | Bastion | 3 December 2010 | 0 | 1 |

**Table 4. Antimicrobial susceptibilities of *Klebsiella variicola* compared with *Klebsiella pneumoniae* isolates, No. (%).**

| Antimicrobial | *K. pneumoniae* susceptibility (N = 221) | *K. variicola* susceptibility (N = 10) | *P*-value |
|---|---|---|---|
| Cefazolin | 50 (22) | 8 (80) | <0.01 |
| Ceftriaxone | 73 (33) | 10 (100) | <0.01 |
| Cefepime | 81 (37) | 10 (100) | <0.01 |
| Levofloxacin | 135 (61) | 9 (90) | 0.09 |
| Piperacillin-tazobactam | 99 (44) | 10 (100) | <0.01 |
| Meropenem | 213 (96) | 10 (100) | 1.00 |
| Amikacin | 200 (90) | 10 (100) | 0.31 |

years (IQR: 21–46), median ISS of 38 (IQR: 30–45), and all were injured by improvised explosive devices. There were no statistically significant differences between these characteristics and those of patients with *K. pneumoniae* infections.

Four of the five *K. variicola* isolates were collected from blood specimens. In comparison, 24 (18%) of 134 *K. pneumoniae* infecting isolates were from blood specimens (p = 0.04). No infecting *K. variicola* isolates were MDR compared to 99 (74%) of infecting *K. pneumoniae* (p<0.01). All *K. variicola* isolates were string test negative.

The PFGE analysis demonstrated eight genetically unique *K. variicola* isolates (Fig 1). The three infecting *K. variicola* isolates identical by PFGE were isolated from one patient's blood cultures on days 28 and 29 post-injury. Three patients with unrelated *K. variicola* isolates were treated in the same Kandahar facility (Afghanistan) over a 12-month period. The remaining two patients with *K. variicola* were treated in the same facility in Bastion province (Southern Afghanistan) 15 months apart, but were unique strains by PFGE.

## Discussion

This study emphasizes the challenging resistance patterns of *K. pneumoniae* infections complicating battlefield trauma, even on initial isolation. It is also the first analysis of a broad archive of *K. pneumoniae* samples to systematically evaluate for *K. variicola* misidentification. Notably, while *K. variicola* infections were more likely to be associated with bacteremia, they were less likely to be MDR.

Gram-negative resistance is thought to develop in response to antimicrobial pressures and our patients are no exception as all had prolonged hospital stays and antimicrobial exposure prior to isolation of MDR *Klebsiella* infections. While there were no statistically significant differences between those with initial MDR compared with non-MDR infections, this is likely due to small numbers of patients. Similar to evidence in this study, evaluation of trauma

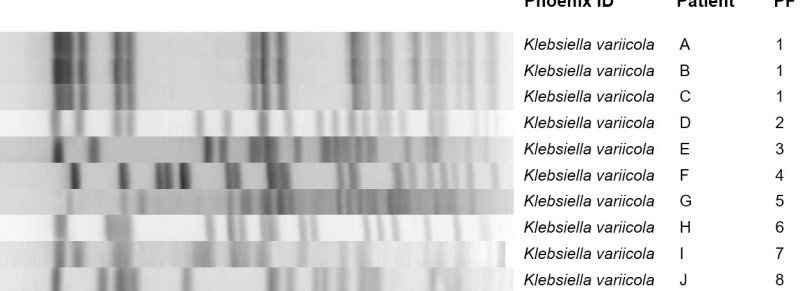

**Fig 1. Pulsed-field gel electrophoresis analysis of 10 *K. variicola* isolates collected from wounded military personnel.**

patients either in developing countries or countries experiencing conflicts have demonstrated high rates of MDR Gram-negative infections [20–22]. For example, civilian trauma patients from the ongoing conflict in Syria were noted to have rates of MDR *K. pneumoniae* infections >80% [23]. Furthermore, nearly 74% of infecting *K. pneumoniae* isolates from a group of South African trauma patients were MDR [20]. These studies, along with others, have not only demonstrated high rates of MDR Gram-negative trauma-associated infections, but also confirmed risk factors for MDR infections to include older age, higher ISS, and prolonged intensive care unit length of stay [24–26]. Our data, which revealed a median of 15 days between injury and *Klebsiella* spp. isolation among these severely injured patients, also emphasizes the nosocomial nature of this pathogen. Adherence to infection control practices and antimicrobial stewardship practices are vital to mitigating the challenge presented by MDR organisms [27–29].

Three important points can be highlighted regarding *K. variicola*. One, this is the first study to systematically evaluate and comprehensively assess rates of misidentification of *K. variicola* in a unique trauma patient population. The overall incidence of *K. variicola* isolates in our battlefield wounded group (4%) is on the lower end of the range that has been described in the literature. In one study, Maatallah and colleagues [7] reported on *K. variicola* BSIs where they noted that 24% of *K. pneumoniae* isolates over a three-year period (2007–2009) were misidentified initially and later correctly identified as *K. variicola*. An outbreak of BSI in neonates in a Bangladesh neonatal intensive care unit also described a high rate of *K. variicola* (38% among 36 bacteremia patients) [6]. Our data identified a relatively small incidence of *K. variicola* (4% [95% CI: 1.5–6.5]), which is likely related to our use of a broader trauma population, rather than focusing on a specific source of infection, like bloodstream isolates. The findings of our study are similar to those described by Rodriguez-Medina and colleagues [8] who examined 1,060 *K. pneumoniae* clinical isolates from a group of hospitals in Mexico and reported a *K. variicola* prevalence of 2.1%. Furthermore, Long and colleagues [30] while examining misidentification in ESBL-producing *K. pneumoniae*, identified nearly 2% as *K. variicola*. The overall proportion of misidentified *K. variicola* bloodstream isolates (4 isolates misidentified; 14% of 28) in our study is comparable to these prior published reports. This could suggest that beyond BSIs, misidentification of *K. variicola* is uncommon; however, further analysis is warranted.

The second notable point is that four of the five infecting *K. variicola* isolates were identified from blood. Even though these are small numbers, coupled with what has been published previously, it does raises important questions. Is there some characteristic of *K. variicola* that leads to an increased propensity to invade the bloodstream? And does this, or some other characteristic portend worse patient outcomes? When Maatallah and colleagues published their work on *K. variicola* BSIs in 2014, the association with *K. variicola* BSI and mortality was alarming (30-day mortality approached 30% compared to 13.5% with *K. pneumoniae*) [7]. Concerns continued to be raised by a study reporting 54% mortality amongst patients who were infected with *K. variicola* in a Bangladesh neonatal intensive care unit [6]. Description of the first hypermucoviscous *K. variicola* organism in 2015 raised further questions that hypermucosviscosity might be an underlying mechanism to explain its invasive behavior [11]. While our findings did show an association between *K. variicola* and BSIs, there was no increase in mortality in these patients and none showed the hypermucoviscosity phenotype. Specifically, no patients with *K. variicola* infections in our analysis died, in contrast to worrisome mortality rates reported previously. It is important to note that the lack of the hypermucoviscous phenotype does not necessarily exclude a particular isolate from the hypervirulent phenotype [31].

The third point is that *K. variicola* isolates in this group were significantly more susceptible to first-line antimicrobials than *K. pneumoniae* isolates (Table 4). Data on susceptibility patterns for *K. variicola* are sparse due to the limited number of studies published; however,

greater susceptibility does seem to be a common emerging theme. In particular, Maatalah and colleagues reported no MDR patterns in 34 *K. variicola* isolates from their repository analysis [7]. Moreover, assessment of *K. pneumoniae* (6 isolates) and *K. variicola* (2 isolates) isolated from a single patient demonstrated that while *K. pneumoniae* was imipenem-resistant, *K. variicola* was susceptible [32]. Our findings provide further evidence that while *K. variicola* may be more likely to be invasive, it is less likely to be MDR.

Our analysis is limited by the small numbers of *Klebsiella* infections and isolates, which may have impacted the ability to detect statistical differences between patients with MDR and non-MDR infection. As suggested by the variation in number of isolates over the years of the study, some of this limitation is likely related to the slowing rate of combat evacuations as the conflicts de-escalated and with the end of combat operations in Afghanistan by the end of 2014. Nevertheless, the findings provide useful information to help with rank ordering likelihood of this specific infection in trauma patients with these characteristics. Our second limitation lies in the PFGE analysis, which necessarily provides a broader view of relatedness among isolated organisms. However in this case, PFGE provides an important first pass analysis for clonality among isolates and argues that similarities between isolates that were associated with the Bastion group of patients, the overlapping location, temporal association of their isolation, and historical challenges with infection control in the deployed environment argues in favor of an outbreak [27,33]. Lastly, the analysis is limited by the lack of granularity in the MDR analysis–for example, further work would be helpful to include mechanisms of resistance such as de-repressed AmpC. The role *K. variicola* plays in healthcare-associated infections and outbreaks has been previously suggested [6,7] and further study of the *Klebsiella* genus over the last four years [8,30] has rapidly expanded the understanding of how the species *variicola* fits within the broader complex.

Overall, our data support findings from previous studies from infections complicating combat-trauma: these patients were severely injured, had prolonged hospital stays with exposure to antibiotics, and *K. pneumoniae* infections were marked by a high rate of multidrug resistance, even early in their hospitalizations [2,4,14,15]. *K. variicola* was more likely to be identified from BSIs and invasive disease. Further work is needed to help clinicians in interpreting the clinical significance of *K. variicola*. Larger *Klebsiella* isolate repositories that could be analyzed for resistance patterns, clinical characteristics, and for misidentification of *K. variicola* would be helpful in this regard. More phylogenetic work expanding our understanding of the genus *Klebsiella* is vital to further characterizing the role of specific species and their relevant pathogenic characteristics.

## Supporting information

**S1 Fig. Raw images for pulsed-field gel electrophoresis analysis of 10 *K. variicola* isolates collected from wounded military personnel.**
(PDF)

## Acknowledgments

We are indebted to the Infectious Disease Clinical Research Program Trauma Infectious Disease Outcomes Study team of clinical coordinators, microbiology technicians, data managers, clinical site managers, and administrative support personnel for their tireless hours to ensure the success of this project.

## Disclaimer

The views expressed are those of the authors and do not reflect the official views of the Uniformed Services University of the Health Sciences, Henry M. Jackson Foundation for the

Advancement of Military Medicine, Inc., the National Institute of Health or the Department of Health and Human Services, Brooke Army Medical Center, Walter Reed National Military Medical Center, Landstuhl Regional Medical Center, the U.S. Army Medical Department, the U.S. Army Office of the Surgeon General, the Department of Defense, or the Departments of the Army, Navy or Air Force. Mention of trade names, commercial products, or organizations does not imply endorsement by the U.S. Government.

## Author Contributions

**Conceptualization:** John L. Kiley, Katrin Mende, Timothy J. Whitman, Joseph L. Petfield, David R. Tribble, Dana M. Blyth.

**Data curation:** Katrin Mende, Miriam L. Beckius, Susan J. Kaiser, Dan Lu.

**Formal analysis:** John L. Kiley, Katrin Mende, Miriam L. Beckius, Susan J. Kaiser.

**Investigation:** John L. Kiley, Katrin Mende, M. Leigh Carson, Dan Lu, Timothy J. Whitman, Joseph L. Petfield, David R. Tribble, Dana M. Blyth.

**Writing – original draft:** John L. Kiley.

**Writing – review & editing:** Katrin Mende, Miriam L. Beckius, Susan J. Kaiser, M. Leigh Carson, Dan Lu, Timothy J. Whitman, Joseph L. Petfield, David R. Tribble, Dana M. Blyth.

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
