## [Decision Letter · Decision Letter 0]

13 May 2021

PONE-D-21-10025

Resistance patterns and clinical outcomes of Klebsiella pneumoniae and invasive Klebsiella variicola in trauma patients

PLOS ONE

Dear Dr. Kiley,

Thank you for submitting your manuscript to PLOS ONE. After careful consideration, we feel that it has merit but does not fully meet PLOS ONE’s publication criteria as it currently stands. Therefore, we invite you to submit a revised version of the manuscript that addresses the points raised during the review process.

A number of questions pertaining to methodology have been raised by the reviewers. Please address all reviewer comments point by point. 

We look forward to receiving your revised manuscript.

Kind regards,

Iddya Karunasagar

Academic Editor

PLOS ONE

Additional Editor Comments:

Two reviewers have commented on the manuscript and raised important questions about the methodology used for antimicrobial sensitivity testing, to characterise hypervirulent strains, differentiation of colonisers from those causing infection, species of Klebsiella considered and other aspects of the manuscript. Please address all points raised by the reviewers and revise the manuscript.

Journal Requirements:

[Support for this work (IDCRP-024) was provided by the Infectious Disease Clinical Research Program (IDCRP), a Department of Defense program executed through the Uniformed Services University of the Health Sciences, Department of Preventive Medicine and Biostatistics through a cooperative agreement with The Henry M. Jackson Foundation for the Advancement of Military Medicine, Inc. (HJF). This project has been funded by the National Institute of Allergy and Infectious Diseases, National Institutes of Health, https://www.niaid.nih.gov/, under Inter-Agency Agreement Y1-AI-5072 to DRT, the Defense Health Program, U.S. DoD, under award HU0001190002 to DRT, the Department of the Navy under the Wounded, Ill, and Injured Program (HU0001-10-1-0014) to DRT, and the Military Infectious Diseases Research Program, https://midrp.amedd.army.mil/ (HU0001-15-2-0045) to KM.  The funders had no role in study design, data collection and analysis, decision to publish, or preparation of the manuscript.].   

We note that one or more of the authors are employed by a commercial company: Advancement of Military Medicine, Inc.

Reviewers' comments:

Reviewer's Responses to Questions

**Comments to the Author**

1. Is the manuscript technically sound, and do the data support the conclusions?

Reviewer #1: Partly

Reviewer #2: Yes

2. Has the statistical analysis been performed appropriately and rigorously? 

Reviewer #1: Yes

Reviewer #2: N/A

3. Have the authors made all data underlying the findings in their manuscript fully available?

Reviewer #1: Yes

Reviewer #2: Yes

4. Is the manuscript presented in an intelligible fashion and written in standard English?

Reviewer #1: Yes

Reviewer #2: Yes

5. Review Comments to the Author

Reviewer #1: The comments are attached. The comments for the paper " Resistance patterns and clinical outcomes of Klebsiella pneumoniae and invasive K. variicola in trauma patients". There are comments and observations made which the authors are requested to address before publication

Reviewer #2: Dear Authors

The paper describes the speciation of Klebsiella variicola spp among previously isolated from trauma patients and identified as Klebsiella pneumoniae along with the resistance patters and clinical outcomes. While epidemiological studies like this throws so much light on speciation and outcomes of infections in patients with trauma especially in army veterans returning from Afghanistan, the authors may consider the following to improve the manuscript.

1. The authors have included isolates that were colonizers and those from infections. However, the time line of the colonization and if they had any clinical significance to go on and cause infection does not seem clear. How long ago from the posting in Afghanistan did they colonize with the isolates? Were these isolates fond before the posting or after the posting? Did the colonizers have any relevance to the isolates in the wound swabs of the patients?

2. Why did the authors stick to identifying only 3 species when Klebsiella pneumoniae , Klebsiella quasipneumoniae subsp. quasipneumoniae , Klebsiella quasipneumoniae subsp. similipneumoniae , Klebsiella variicola subsp. variicola , Klebsiella variicola subsp. tropica , Klebsiella quasivariicola , Klebsiella africana have been described?

3. Would the authors opine that these infections and colonizations were health care associated/ health care acquired infections?

4. Why was PCR not used to differentiated the hipervirulent isolates? String test seems to be a very crude method. Further, data on hypermucoviscosity vs hypervirulence may be a great add on.

5. What were the number of ESBL's detected in this population of isolates studied? What was the method employed to classify them as ESBL's? Which CLSI guideline was used ( Version)

6. Isolates from clinical samples need to be detailed if they were associated with healthcare associated infection. Were wound swabs the main source. If so were the wound sites in the groin region? If so were the same isolates found as colonizers and then went on to cause infections? The percentages of isolates and their distribution may also be shown in a figure. However, the percentage of K. variicola seems too small to draw any conclusions especially when isolated from the blood culture

7. Since the study period is between 2009-2014, in 5 years were there changes in the isolation rates? Maybe a timeline of the distribution of isolates in a figure would be useful?

6. PLOS authors have the option to publish the peer review history of their article (what does this mean?). If published, this will include your full peer review and any attached files.

Reviewer #1: No

Reviewer #2: No

---

## [Author Response · Author response to Decision Letter 0]

14 Jun 2021

Additional Editor Comments:

Two reviewers have commented on the manuscript and raised important questions about the methodology used for antimicrobial sensitivity testing, to characterise hypervirulent strains, differentiation of colonisers from those causing infection, species of Klebsiella considered and other aspects of the manuscript. Please address all points raised by the reviewers and revise the manuscript.

Author Response: Thank you for the opportunity to revise. We have reviewed and comments from the reviewers and made changes to the manuscript accordingly. 

Journal Requirements:

Author Response: The manuscript has been formatted according to the style requirements and named according to the file naming guidelines. 

Author Response: The original uncropped and unadjusted image underlying the gel results reported in the manuscript is included in the Supporting Information file uploaded with the manuscript. 

[Support for this work (IDCRP-024) was provided by the Infectious Disease Clinical Research Program (IDCRP), a Department of Defense program executed through the Uniformed Services University of the Health Sciences, Department of Preventive Medicine and Biostatistics through a cooperative agreement with The Henry M. Jackson Foundation for the Advancement of Military Medicine, Inc. (HJF). This project has been funded by the National Institute of Allergy and Infectious Diseases, National Institutes of Health, https://www.niaid.nih.gov/, under Inter-Agency Agreement Y1-AI-5072 to DRT, the Defense Health Program, U.S. DoD, under award HU0001190002 to DRT, the Department of the Navy under the Wounded, Ill, and Injured Program (HU0001-10-1-0014) to DRT, and the Military Infectious Diseases Research Program, https://midrp.amedd.army.mil/ (HU0001-15-2-0045) to KM. The funders had no role in study design, data collection and analysis, decision to publish, or preparation of the manuscript.]. 

We note that one or more of the authors are employed by a commercial company: Advancement of Military Medicine, Inc.

Author Response: The company you are referencing is the Henry M. Jackson Foundation for the Advancement of Military Medicine, Inc., which is a non-profit authorized by Congress to support research at the Uniformed Services University. The Henry M. Jackson Foundation for the Advancement of Military Medicine, Inc., is directly referenced in the submitted Funding Statement that was supplied during the initial submission “Support for this work (IDCRP-024) was provided by the Infectious Disease Clinical Research Program (IDCRP), a Department of Defense program executed through the Uniformed Services University of the Health Sciences, Department of Preventive Medicine and Biostatistics through a cooperative agreement with The Henry M. Jackson Foundation for the Advancement of Military Medicine, Inc. (HJF).” The text has been revised to reference authors who received salaries from HJF. “Support in the form of salaries was provided by HJF for authors KM, SJK, MLC, and DL; HJF did not have any additional role in the study design, data collection and analysis, decision to publish, or preparation of the manuscript. The specific roles of these authors are articulated in the ‘author contributions’ section.” 

We reviewed the author contribution section and no changes are needed.

Author Response: Please see response above

Author Response: The Competing Interests Statement has been revised to be the following: “KM, SJK, MLC, and DL are employees of the Henry M. Jackson Foundation for the Advancement of Military Medicine, Inc. (HJF), a not-for-profit Foundation authorized by Congress to support research at the Uniformed Services University of the Health Sciences (USU) and throughout military medicine. This does not alter our adherence to PLOS ONE policies on sharing data and materials. Please see Data Availability Statement.”

Author Response: Please see response above

Author Response: The full revised text of the Author Disclosure (funding) statement and Competing Interest Statement is included in the cover letter.

 

Reviewers' comments:

Reviewer's Responses to Questions

Comments to the Author

1. Is the manuscript technically sound, and do the data support the conclusions?

Reviewer #1: Partly

Reviewer #2: Yes

2. Has the statistical analysis been performed appropriately and rigorously?

Reviewer #1: Yes

Reviewer #2: N/A

3. Have the authors made all data underlying the findings in their manuscript fully available?

Reviewer #1: Yes

Reviewer #2: Yes

4. Is the manuscript presented in an intelligible fashion and written in standard English?

Reviewer #1: Yes

Reviewer #2: Yes

5. Review Comments to the Author

Reviewer #1: The comments are attached. The comments for the paper " Resistance patterns and clinical outcomes of Klebsiella pneumoniae and invasive K. variicola in trauma patients". There are comments and observations made which the authors are requested to address before publication

An unique article detailing the incidence of infections and colonization by Klebsiella species, particularly K. variicola which may be misidentified as K. pneumoniae. The authors have selected trauma patients among the army veterans who participated in the battle in Afghanistan and were supposedly hospitalized and underwent procedures for various indications 

The article is well presented and thoughts have been collated to bring out the said manuscript. The following observations are in order: 

1. Colonisation was defined through the study to reflect those isolates obtained from groin surveillance swabs. Any particular reason for doing so, as Klebsiella spp are known to colonise a number of body sites including the respiratory and urinary tract without causing clinical infections. Can the authors substantiate this criteria for selection site for surveillance of colonization 

Author Response: Thank you for the comment. As you rightly point out, Klebsiella spp. are known to colonize many organ systems. As part of infection control measures to identify patients with skin colonization of a wide variety of organisms and to limit transmission of multidrug-resistant organisms, surveillance cultures were collected at admission to the military hospitals (groin/axillary swabs at Landstuhl and groin/axillary/nares swabs at U.S.-based hospitals). The collection of the admission surveillance swabs was per policies at each of the sites and not based on instructions from TIDOS investigators. The sentence defining colonization in the Methods text (lines 93-95) was revised to clarify this point and now reads “Colonization was defined as recovery of isolate from groin swabs obtained as part of targeted infection control surveillance at hospital admission for their deployment-related injury.” 

2. Klebsiella isolate analysis: The authors have alluded to the fact that Vitek and BD systems are unable to differentiate between K. pneumoniae, variicola and quasipneumoniae. Why have the authors chosen just three species for ability or inability of the systems for species differentiation. What may be the reason for this specific focus on the said three species and not the others 

Author Response: Thank you for the comment. At the time of the study development, we had elected to utilize the PCR technique described by Garza-Ramos and colleagues (BMC Microbiol. 2015; 15:64; BMC Microbiol. 2016; 16:43) that would be able to distinguish these three species, primarily because of the ability to identify K. variicola. This decision hinged on our specific clinical question raised by the Sweden bacteraemia data (Maatallah et al. 2014. PLoS One. 9(11): e113539) and further discussed/elucidated by Garza-Ramos and colleagues. Since the laboratory bench work on our project was completed, there has continued to be literature published in the area of the phylogenetic and epidemiologic nature of the Klebsiella genus, leading ultimately to the elucidation of five different species (including here K. quasivariicola). Given this evolution in the landscape of the genus, we added a new reference as #9 (Barrios-Camacho H, et al. Sci Rep. 2019; 9(1): 10610) to the Introduction and clarifying language to the discussion of this manuscript on page 17 (lines 336-338) to give the reader a greater sense of the scope of this work and recent developments that have taken place in the scientific community. 

Does literature suggest that there aren’t other species of Klebsiella that could prove pathogenic other than the ones tested for here in this article? 

Author Response: Thank you for the comment. There has been literature published (or is in pre-print) since we completed our study that indicates the occurrence of other species of Klebsiella that have specific pathogenicity associated with them, in particular the association with K. variicola as a uropathogen and hypervirulent strains of K. quasipneumoniae. The recently described K. africanensis and K. quasivariicola would also deserve attention for future work attempting to further fill in the details of the pathogenicity of the entire complex. As above, we have alluded to this evolving information with edits in lines 336-338.

3. The same (? bias) is reflected in this subsequent section on PFGE of the isolates where the PCR method and the primers used were dictated by the choice of the species rather than a broad range of species. It may have been useful to carry out the PCR to be able to identify other unusual species of Klebsiella too. Can this be substantiated by the authors? 

Author Response: We agree that broad range PCR would have given additional data. Based off what we knew from the published literature at the time of the initiation of the analysis, we felt a more targeted approach, particularly of a clinical cohort of patients would be better suited to try to answer the specific question we had on K. variicola. 

4. Virulence factors for Hypervirulence in Klebsiella species is best done with a genetic analysis of the isolates where there is an attempt made to amplify the presence of genes such as mag A gene ( for instance) String test is a crude method and is not acceptable for drawing conclusion on the possible hypervirulence of Klebsiella species . 

Author Response: We completely agree that the string test is a crude test and clarified the text in the Methods (the choice of using this bench side test; page 6, lines 131-133), as well as in the Discussion on page 16 (lines 311-312) to emphasize the careful distinctions made between virulence and hyperviscosity made by Catalan-Najera and colleagues in Virulence (2017; 8[7]:1111-1123) regarding this important point. 

5. Of the 51 infecting Klebsiella isolates, 16 were from the respiratory tract. Can the authors furnish information/ data on whether these respiratory infections were Lower respiratory tract infections, VAP, HAP etc. This is important as Klebsiella is known to colonise the respiratory tract of hospital and facility in- patients. The same holds good for the 5 isolates from the urinary tract. 

Author Response: Thank you for the question. These were lower respiratory tract infections – the way the initial data collection was done, the isolates were definitional not colonizers. Timing of infections was also part of the data collection, but not explicitly identified as VAP. Urinary isolates were considered associated with infection as defined in the methods (lines 95-96). We have since clarified the text in the Methods (lines 93-96) to reflect that only isolates collected from groin swabs were considered colonizers, and all remaining clinical isolates included were considered infecting.

6. Isolate analysis: The fourth line in the 2nd paragraph talks about substantial resistance to ESBL inhibitors. Can the authors qualify this please, as one only alludes to Beta lactam lactamase inhibitors in scientific literature. The only inhibitor visible in Table -4 was Piperacillin tazobactam and this is not classified scientifically as an ESBL inhibitor 

Author Response: Thank you for the comment and we agree. We have updated the language on page 12 (line 225) to specify piperacillin-tazobactam.

7. There was a mention in the Materials and methods section of classifying isolates as MDR based on resistance / ESBL/ Carbapenamase etc. However the same is not reflected in the Results section as to how many of the isolates showed ESBL, derepressed Amp C or a Carbapenamase enzyme. This may be an useful information in a manuscript dealing with resistance patterns and related clinical outcomes 

Author Response: Thank you for your comment and we agree. Ultimately, we had very few CRE organisms, and we did not have the bench work to support genetic evidence for specific cases of suspected Amp C derepression, so we felt that for comparing groups, our analysis would be strongest by grouping the mechanisms into multidrug resistance. We have included text on page 17 (lines 334-335), fully acknowledging this limitation. 

8. Klebsiella variicola: in the Results section: The authors have stated that 80% of the K. variicola isolates were from blood cultures. 4 of the 5 isolates does not qualify for a percentage. The same concept is again represented in the discussion section.

Author Response: Thank you for your comment. We have revised the sentence in the Results to remove the 80%, so it is a statement of numbers only. The statement in the Discussion was revised to be the following: “The second notable point is that four of the five infecting K. variicola isolates were identified from blood.” 

We also adjusted a sentence in the abstract that referenced the data to now be “Compared to K. pneumoniae, infecting K. variicola isolates were more likely to be from blood (4/5 versus 24/134, p=0.04), and less likely to be multidrug-resistant (0/5 versus 99/134, p<0.01).” 

9. Discussion section Page 14: The last two lines taken about the “overall proportion of misidentified blood stream isolates 14% of 28 in our study”. this statement needs clarification as this does impact the discussion and the concluding paragraph. 

Author Response: Thank you for your question. There were 28 BSI isolates initially identified as K. pneumoniae, but 4 were later classified as K. variicola. The text has been revised for clarity to read “Nevertheless, the overall proportion of misidentified K. variicola bloodstream isolates (4 isolates misidentified; 14% of 28) in our study is comparable to these prior published reports.” The goal of this language was to compare our reported misidentification in bloodstream isolates to the other published reports. 

Reviewer #2: Dear Authors

The paper describes the speciation of Klebsiella variicola spp among previously isolated from trauma patients and identified as Klebsiella pneumoniae along with the resistance patters and clinical outcomes. While epidemiological studies like this throws so much light on speciation and outcomes of infections in patients with trauma especially in army veterans returning from Afghanistan, the authors may consider the following to improve the manuscript.

Author Response: Thank you for your comments

1. The authors have included isolates that were colonizers and those from infections. However, the time line of the colonization and if they had any clinical significance to go on and cause infection does not seem clear. How long ago from the posting in Afghanistan did they colonize with the isolates? Were these isolates fond before the posting or after the posting? Did the colonizers have any relevance to the isolates in the wound swabs of the patients?

Author Response: Thank you for your comments. The colonizing isolates were recovered as part of infection control procedures (groin surveillance swabs) at hospital admission for their deployment-related injury following medical evacuation from Afghanistan. The sentence defining colonization in the Methods text (lines 93-95) was revised to clarify this point and now reads “Colonization was defined as recovery of isolates from groin swabs obtained as part of targeted infection control surveillance at hospital admission for their deployment-related injury.” In addition, the sentence (lines 110-111) that described the linking of infecting isolates to colonizing isolates was revised to “All colonizing isolates linked with infecting isolates (defined as isolation from groin admission swab prior to infection) were included.” 

2. Why did the authors stick to identifying only 3 species when Klebsiella pneumoniae, Klebsiella quasipneumoniae subsp. quasipneumoniae, Klebsiella quasipneumoniae subsp. similipneumoniae, Klebsiella variicola subsp. variicola, Klebsiella variicola subsp. tropica, Klebsiella quasivariicola, Klebsiella africana have been described?

Author Response: Thank you for the comment. Essentially, we felt that one of the questions we had pre-specified before doing this work was to focus on the role of K. variicola that had been published in the literature and that our N and subsequent design of the bench work didn’t allow us to broadly PCR for the rest of the species within the genus –as you rightly point out. Please also see the response to comment #2 from Reviewer #1.

3. Would the authors opine that these infections and colonizations were health care associated/ health care acquired infections?

Author Response: Thank you for the question—we have added a sentence that clarifies the nosocomial nature of these isolates within the Discussion (lines 274-276), prior to a sentence that discusses the essential role of infection prevention and control as well as antimicrobial stewardship in decreasing their impact.

4. Why was PCR not used to differentiated the hipervirulent isolates? String test seems to be a very crude method. Further, data on hypermucoviscosity vs hypervirulence may be a great add on.

Author Response: Thank you for your comment and we completely agree. We felt that this would be a question that many readers might have (the string test), but ultimately after this crude screening test, felt that beyond reporting the data, there were not any additional conclusions to be drawn. A statement was added to the Discussion on page 16 (lines 311-312), which states “It is important to note here that the lack of the hypermucoviscous phenotype does not necessarily exclude a particular isolate from the hypervirulent phenotype” and as above, added the reference by Catalan-Najera and colleagues in Virulence (2017; 8[7]:1111-1123) regarding this important point. 

5. What were the number of ESBL's detected in this population of isolates studied? What was the method employed to classify them as ESBL's? Which CLSI guideline was used ( Version)

Author Response: ESBL’s were not explicitly re-identified outside of the automated breakpoint data that accompanies the BD Phoenix. The CLSI guideline used was M100, 28th edition published in 2018 and that information was added to line 119 on page 6 in the Methods section. 

6. Isolates from clinical samples need to be detailed if they were associated with healthcare associated infection. Were wound swabs the main source. If so were the wound sites in the groin region? If so were the same isolates found as colonizers and then went on to cause infections? The percentages of isolates and their distribution may also be shown in a figure. However, the percentage of K. variicola seems too small to draw any conclusions especially when isolated from the blood culture

Author Response: Thank you for your comments. The text in the Results Isolate Analysis section was revised to clarify the number of K. pneumoniae infecting isolate sources. The revised sentence reads “Sources of the 134 K. pneumoniae infecting isolates were wound (N=57, 43%), respiratory (N=29, 22%), blood (N=24, 18%), intraabdominal (N=4, 3%), and other (N=20, 15%).” Text in the Klebsiella variicola section was also revised to clarify the sources of the infecting isolates and now reads “Five isolates were colonizers from groin swabs, while the five isolates associated with infections were from blood (N=4), and intraabdominal specimens (N=1).”

Yes, wound swabs were the main source for K. pneumoniae, but it was blood for K. variicola. Wound sites could derive from the abdomen, gluteal, hand, lower leg, pelvis, thigh, and upper arm. Additionally, we further clarified within the methods (as discussed in response to reviewer 1) the definitions of colonizing vs infecting isolates. 

7. Since the study period is between 2009-2014, in 5 years were there changes in the isolation rates? Maybe a timeline of the distribution of isolates in a figure would be useful?

Author Response: Thank you for the comment. We did have similar questions about isolation rates, particularly when it comes out in outbreak analyses; however, outside of our outbreak analysis where there were important features of timing, we thought that battlefield kinetics and timing of injuries would be hard to control for. Text related to the number of isolates collected over the study years was added to the Results on page 11, lines 214-219. The new text states that ‘The years of K. pneumoniae collection were 28 (13% of 221) isolates in 2009, 95 (43%) in 2010, 30 (14%) in 2011, 51 (23%) in 2012, 13 (6%) in 2013, and 4 (2%) in 2014. There were no K. variicola isolates collected in 2009, while three were collected in 2010, four were collected in 2011, and one isolate was collected per year from 2012-2014. Four of the K. quasipneumoniae isolates were collected in 2011, one isolate in 2013, and one isolate in 2014.’ 

We have also added a sentence to the Discussion (lines 324-326) to discuss how the number of isolates tapering off likely reflects the decreasing numbers of evacuated casualties towards the end of the study period, coinciding with the end of combat operations in Afghanistan.

---

## [Decision Letter · Decision Letter 1]

21 Jul 2021

Resistance patterns and clinical outcomes of Klebsiella pneumoniae and invasive Klebsiella variicola in trauma patients

PONE-D-21-10025R1

Dear Dr. Kiley,

We’re pleased to inform you that your manuscript has been judged scientifically suitable for publication and will be formally accepted for publication once it meets all outstanding technical requirements.

Kind regards,

Iddya Karunasagar

Academic Editor

PLOS ONE

Additional Editor Comments (optional):

All reviewer comments have been addressed satisfactorily.

Reviewers' comments:

Reviewer's Responses to Questions

**Comments to the Author**

1. If the authors have adequately addressed your comments raised in a previous round of review and you feel that this manuscript is now acceptable for publication, you may indicate that here to bypass the “Comments to the Author” section, enter your conflict of interest statement in the “Confidential to Editor” section, and submit your "Accept" recommendation.

Reviewer #1: All comments have been addressed

Reviewer #2: All comments have been addressed

2. Is the manuscript technically sound, and do the data support the conclusions?

Reviewer #1: Yes

Reviewer #2: Yes

3. Has the statistical analysis been performed appropriately and rigorously? 

Reviewer #1: N/A

Reviewer #2: N/A

4. Have the authors made all data underlying the findings in their manuscript fully available?

Reviewer #1: Yes

Reviewer #2: Yes

5. Is the manuscript presented in an intelligible fashion and written in standard English?

Reviewer #1: Yes

Reviewer #2: Yes

6. Review Comments to the Author

Reviewer #1: All previous queries raised have been adequately addressed . All specific questions have been answered by the authors

Reviewer #2: Dear authors.

All comments have been adequately addressed and adequate references have been added to substantiate the explanation.

7. PLOS authors have the option to publish the peer review history of their article (what does this mean?). If published, this will include your full peer review and any attached files.

Reviewer #1: No

Reviewer #2: **Yes: **Anusha Rohit

---

## [Editor Report · Acceptance letter]

23 Jul 2021

PONE-D-21-10025R1 

Resistance patterns and clinical outcomes of *Klebsiella pneumoniae* and invasive *Klebsiella variicola* in trauma patients 

Dear Dr. Kiley:

I'm pleased to inform you that your manuscript has been deemed suitable for publication in PLOS ONE. Congratulations! Your manuscript is now with our production department. 

Kind regards, 

on behalf of

Dr. Iddya Karunasagar 

Academic Editor

PLOS ONE